# Remote Sensing of Maize Plant Height at Different Growth Stages Using UAV-Based Digital Surface Models (DSM)

Leon Hinrich Oehme [1,*,†], Alice-Jacqueline Reineke [1,†], Thea Mi Weiß [2], Tobias Würschum [2], Xiongkui He [3] and Joachim Müller [1]

1   Tropics and Subtropics Group, Institute of Agricultural Engineering, University of Hohenheim,
    Garbenstraße 9, 70599 Stuttgart, Germany; a.reineke@uni-hohenheim.de (A.-J.R.);
    joachim.mueller@uni-hohenheim.de (J.M.)

2   Institute of Plant Breeding, Seed Science and Population Genetics, University of Hohenheim,
    Fruwirthstraße 21, 70599 Stuttgart, Germany; theami.weiss@uni-hohenheim.de (T.M.W.);
    tobias.wuerschum@uni-hohenheim.de (T.W.)

3   College of Science, China Agricultural University, Beijing 100193, China; xiongkui@cau.edu.cn

*   Correspondence: leon.oehme@uni-hohenheim.de

†   These authors contributed equally to this work.

**Abstract:** Plant height of maize is related to lodging resistance and yield and is highly heritable but also polygenic, and thus is an important trait in maize breeding. Various manual methods exist to determine the plant height of maize, yet they are labor-intensive and time consuming. Therefore, we established digital surface models (DSM) based on RGB-images captured by an unmanned aerial vehicle (UAV) at five different dates throughout the growth period to rapidly estimate plant height of 400 maize genotypes. The UAV-based estimation of plant height ($PH_{UAV}$) was compared to the manual measurement from the ground to the highest leaf ($PH_L$), to the tip of the manually straightened highest leaf ($PH_S$) and, on the final date, to the top of the tassel ($PH_T$). The best results were obtained for estimating both $PH_L$ ($0.44 \leq R^2 \leq 0.51$) and $PH_S$ ($0.50 \leq R^2 \leq 0.61$) from 39 to 68 days after sowing (DAS). After calibration the mean absolute percentage error (MAPE) between $PH_{UAV}$ and $PH_S$ was in a range from 12.07% to 19.62%. It is recommended to apply UAV-based maize height estimation from 0.2 m average plant height to maturity before the plants start to senesce and change the leaf color.

**Keywords:** plant height; unmanned aerial vehicle (UAV); maize; high-throughput phenotyping; digital surface model (DSM); photogrammetry; drone

## 1. Introduction

One important trait of breeding programs in maize (*Zea mays* L.) is plant height, as it can be a crucial factor for yield potential [1]. Plant height is also strongly related to lodging resistance [2,3] and is highly heritable despite its complicated polygenic nature [4]. Plant height measured in early stages, such as the four-leaf stage (V4), is an important tool for the assessment of early plant development in maize [5]. An early and fast establishment of the maize canopy is crucial in temperate latitudes due to the limitation of solar radiation [6]. Besides breeding programs, early plant height measurement could also be beneficial for farmers to predict the final grain yield in maize [7].

Various manual methods exist to determine the plant height of maize, which are mainly dependent on the stage when the maize is measured. One common method is to straighten the youngest leaf's tip to determine the distance to the ground [8]. After flowering, Peiffer et al. [4] measured the distance from ground to the flag leaf base to obtain results independent of tassel length. Another method is to simply measure the highest point of the plant, which can be the youngest leaf or the tassel in later developmental stages [3,9]. All these methods produce different results, are labor-intensive as well as

time-consuming in typically large breeding trials, and can be especially challenging for very tall plants, which highlights the need for more efficient measurement methods.

Manual methods to estimate plant height can be replaced by more efficient approaches based on unmanned aerial vehicles (UAV) [9–14]. Different sensors attached to the UAV enable creating digital surface models (DSM) that depict the topography of the entire field's surface comprising crop canopy and visible ground. One way to compute DSM is to use data from laser-emitting LiDAR (light detection and ranging) sensors. By measuring the time taken for the laser to travel from the UAV to the object and back, object heights can be estimated precisely. However, LiDAR sensors are expensive and limited to the purpose of measuring distances [14]. Another way to create DSM is based on RGB sensors, which are cheaper and more versatile than LiDAR sensors. By taking overlapping images of an object from different angles, DSM are created via photogrammetry [15].

RGB-based DSM for the estimation of plant height has been successfully tested for various crops, but in many cases, plants are only surveyed in later developmental stages. Su et al. [9] estimated the plant height of maize once in the jointing stage and once in the milk stage. Both Watanabe et al. 2017 [13], who surveyed sorghum, and Gilliot et al. [10], who surveyed maize, estimated plant height after anthesis. Hassan et al. [11] presented data in the booting and mid-grain fill stage of wheat. Thus, there is a research gap for investigating UAV-based plant height estimation in juvenile stages and for identifying the limitations of UAV-based plant height estimation dependent on the growth stage.

In this study, plant height of maize was estimated by DSM based on RGB data obtained by UAV. To this end, a field trial was surveyed five times during the growing season, not only to evaluate the estimation of plant height by DSM in general but also to compare estimation accuracy at different dates and developmental stages. In particular, our objectives were to (i) identify the optimal period for UAV-based plant height estimation and to (ii) evaluate the accuracy of the approach in estimating maize plant height compared to manual methods.

## 2. Materials and Methods

### 2.1. Field Trial

The field trial was conducted at an experimental farm of the University of Hohenheim, Germany (48°43′05.7″ N, 9°11′20.8″ E) in 2020. Overall, 400 plots representing 400 different maize genotypes of doubled-haploid lines were included in this study. The plot size was $4 \times 1.5$ m, with a sowing density of 8.66 plants/m$^2$. An overview of the field trial can be seen in Figure 1a and the field elevation of the relevant plots is shown in Figure 1b. The field design and germplasm were described in detail in a previous study [16] and the weather conditions during the UAV flights are shown in Table 1. The maize was sown on APR 30 and plant height was measured by UAV (PH$_{UAV}$) as well as manually five times during the growth period at 34, 39, 47, 68, and 120 days after sowing (DAS). The following three manual measurement methods were applied: (i) a meterstick was used to measure the distance from the ground to the highest point of the highest leaf in its natural position (PH$_L$); (ii) the distance between ground and the leaf tip, that represented the highest point of the plant by manually straightening it, was measured (PH$_S$) [8]. According to the growth stage, PH$_L$ and PH$_S$ were measured on the first four dates while (iii) the distance from ground to tassel tip (PH$_T$) was measured on DAS 120. For each date, method, and plot, three representative plants were chosen at random, measured, and averaged. Dealing with doubled-haploid landraces and elite lines in this trial, a very high level of homogeneity can be assumed. Measuring three plants per genotype for the trait plant height is therefore deemed sufficient and a previous study with the same methodology has shown a high broad-sense heritability of 0.91 [8,16]. Since the manual measurements were conducted separately, the plants chosen for PH$_L$ and PH$_S$ might differ.

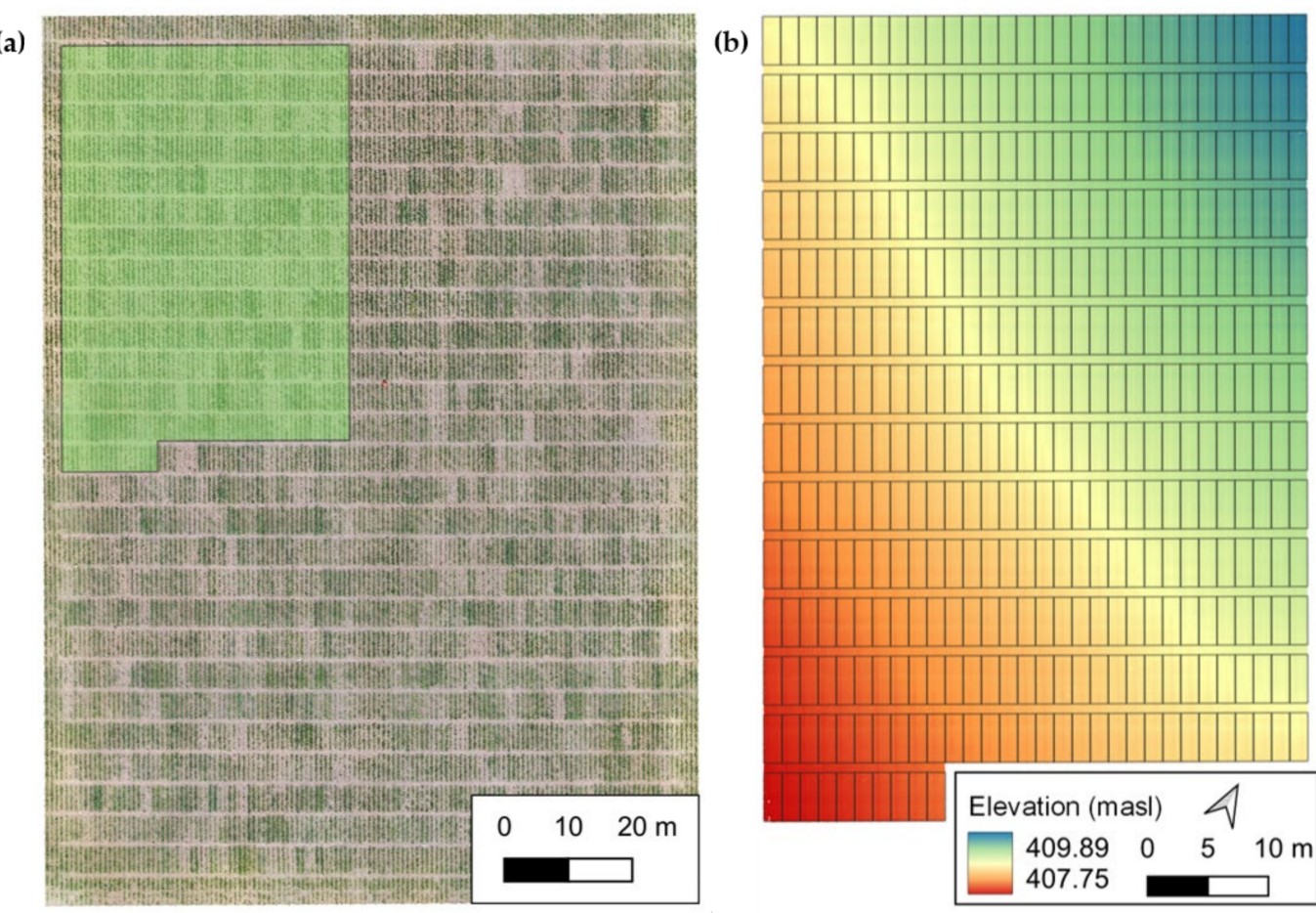

**Figure 1.** Overview of the maize field trial for UAV-based plant height estimation. (**a**) RGB orthomosaic with the 400 relevant plots highlighted in green; (**b**) soil elevation of the 400 plots as received from the flight on DAS 34. The arrow indicates the north; masl: Meter above sea level.

**Table 1.** Weather conditions during the UAV flights; DAS: days after sowing.

| DAS | Date | Time Interval | Temperature (°C) | Rel. Humidity (%) | Wind Speed (m/s) | Global Radiation (W/m$^2$) | Sky Condition |
|---|---|---|---|---|---|---|---|
| 34 | 03 JUNE | 11:57–13:22 | 23.5 | 38.1 | 2.03 | 487 | partly cloudy |
| 39 | 08 [1] JUNE | 15:04–15:26 | 18.0 | 51.8 | 1.76 | 575 | mostly cloudy |
|  |  | 18:29–19:06 | 16.7 | 55.5 | 1.87 | 78 | mostly cloudy |
| 47 | 16 JUNE | 13:16–15:59 | 17.7 | 85.0 | 1.54 | 293 | cloudy |
| 68 | 07 JULY | 11:38–12:42 | 19.1 | 45.2 | 1.37 | 674 | mostly sunny |
| 120 | 28 AUG | 09:49–10:52 | 22.0 | 56.7 | 1.51 | 432 | sunny |

[1] Due to technical issues, the image acquisition on June 8 was completed after an interruption of approximately three hours.

### 2.2. UAV Image Acquisition

The initial step of UAV-based plant height estimation was the image acquisition (Figure 2). A commercial quadcopter (DJI Mavic 2 Pro, Da-Jiang Innovations Science and Technology Co., Ltd., Shenzhen, China) was used as UAV to capture the RGB images. The UAV was equipped with a digital camera (Hasselblad L1D-20c, Victor Hasselblad AB, Gothenburg, Sweden) containing a one-inch complementary metal oxide semiconductor (CMOS) sensor with a resolution capability of 5472 × 3648 pixel (20 mega pixel). The UAV was controlled using the Pix4Dcapture application installed on the DJI smart controller.

Data acquisition was performed in JPEG format containing digital positioning system (GPS) information at 20 m above ground level, which resulted in an average ground sample distance of 0.0049 m (distance between pixel centers). The flight plan consisted of a double grid with 80% front and side overlap with a camera angle of 70° to the horizontal downwards. The average flying speed was 1.5 m/s.

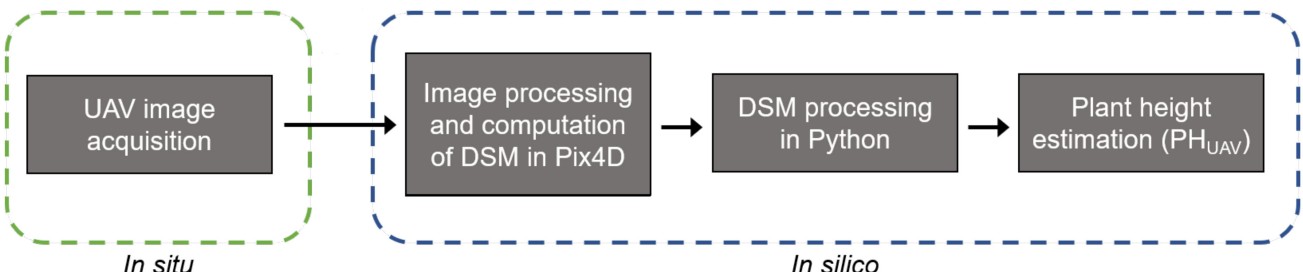

**Figure 2.** General overview of the methods applied for UAV-based plant height estimation; DSM: digital surface model.

### 2.3. UAV Image Preprocessing

Image preprocessing was performed using Pix4DMapper software (version 4.5.6, Pix4D S.A., Prilly, Switzerland) (Figure 2). The following describes the chosen settings; if a setting is not explicitly mentioned, it was not modified from the default settings in the template '3D Maps'. In the step 'Initial processing', the full image scale was used for keypoint extraction. Image pairs were matched using the advanced condition 'Aerial grid or corridor' and geometrically verified. In the calibration settings, the rematch option was enabled. In the step 'Point cloud and mesh', the minimum number of matches was set to 4 and in the advanced options a matching window of 9 × 9 pixel was chosen. In the last step 'DSM, orthomosaic and index' the resolution was set to 1 × GSD. Noise filtering and sharp surface smoothing was applied. The raster DSM was created using the inverse distance weighting algorithm. DSM and orthomosaics were exported in the file format GeoTIFF as merged tiles. Seven black and white checkered targets (0.4 m × 0.4 m), three oil drums (0.6 m, 0.6 m and 0.9 m height), and a soil sensor measuring station served as ground control points to increase the GPS and DSM accuracy. In the last mission, one ground control point (black and white target) was removed in order not to disturb the harvesting process. The GPS information of the ground control points was measured using a GNSS receiver (Stonex S9, Open Port GmbH, Nienburg, Germany). During image processing two of these ground control points (small oil drum, black and white checkered target) were defined as checkpoints in Pix4DMapper.

### 2.4. Procedure of Plant Height Estimation

The generated DSM layers were further processed plot-wise in Python (3.8) [17] (Figure 2) by means of the GDAL (3.3) [18] and OpenCV package (4.5) [19]. First, for basic segmentation, the Otsu threshold [20] was applied to every plot's DSM to classify pixel of high altitude as 'plant' and pixel of low altitude as 'soil'. Subsequently the standard deviation and mean was calculated for both the plant class ($P_{STD}$, $P_{MEAN}$) and the soil class ($S_{STD}$, $S_{MEAN}$). The maximum of the plant class was defined as $P_{MAX}$ and the minimum altitude of the soil class was defined as $S_{MIN}$, which also represent the extreme values of the entire plot's DSM. To measure the distance from the highest point of the plants to the ground without including possible noise or outliers of the DSM, a custom algorithm was developed (Algorithm 1). Depending on which value was lower, the estimated plant height ($PH_{UAV}$) was either based on the classes means and standard deviations ($PH_{STD}$) or the extreme values ($PH_{MINMAX}$). Thus, $PH_{UAV}$ was set to $PH_{STD}$ if exceptional outliers were present in the DSM and set to $PH_{MINMAX}$ in normal cases. Data from one plot (Plot ID 2059)

was excluded from analysis at DAS 68 and from all dates for calibration due to unusual DSM results, which represent an artifact, possibly caused by a bird.

---

**Algorithm 1**: Pseudo-code for estimating plant height ($PH_{UAV}$).

---

**Input:** $S_{MIN}$, $S_{STD}$, $S_{MEAN}$, $P_{MAX}$, $P_{STD}$, $P_{MEAN}$
**set** $PH_{STD}$ **to** $(P_{MEAN} + 5\,P_{STD}) - (S_{MEAN} - 5\,S_{STD})$
**set** $PH_{MINMAX}$ **to** $P_{MAX} - S_{MIN}$
**If** $PH_{STD} < PH_{MINMAX}$ **then**
**set** $PH_{UAV}$ **to** $PH_{STD}$
**Else then**
**set** $PH_{UAV}$ **to** $PH_{MINMAX}$
**Output:** $PH_{UAV}$

---

*2.5. Statistical Analysis*

To assess the accuracy of $PH_{UAV}$, it was compared to the manual measurements $PH_L$ and $PH_S$ on the first four dates and compared to $PH_T$ on the final date. In addition, the relationship between the two manual methods $PH_L$ and $PH_S$ was studied on the first four dates. All statistical analyses were conducted in Python [17]. The Pearson correlation coefficient was calculated by the Scipy package (1.7) [21]. The root mean square error (RMSE), mean absolute percentage error (MAPE), coefficient of determination ($R^2$) were computed by the Scikit-learn package (1.0) [22] and are defined as:

$$RMSE = \sqrt{\frac{1}{n}\sum_{i=1}^{n}(PH_{x,i} - PH_{y,i})^2}, \tag{1}$$

$$MAPE = \frac{1}{n}\sum_{i=1}^{n}\left|\frac{PH_{x,i} - PH_{y,i}}{PH_{x,i}}\right| \tag{2}$$

and

$$R^2 = 1 - \frac{\sum_{i=1}^{n}(PH_{x,i} - PH_{\hat{y},i})}{\sum_{i=1}^{n}(PH_{x,i} - \overline{PH_x})} \tag{3}$$

where $n$ is the number of plots and $PH_{x,i}$ and $PH_{y,i}$ are the plant heights at the $i$-th plot as measured by the methods $x$ and $y$, respectively. $PH_{\hat{y},i}$ is the predicted plant height by the regression model for method $y$ at the $i$-th plot and $\overline{PH_x}$ is the average plant height of all plots as measured by method $x$. The Scikit-learn package was also used to calculate the linear regression and the 70/30 train-validation split (seed = 0) of the plots for calibration. Data visualization was done by the Matplotlib package (3.5) [23].

**3. Results**

*3.1. Plant Development throughout the Measurement Period*

Since the juvenile stage is crucial for plant development, the plants were measured manually and by UAV more frequently in the beginning of the growth period (three dates in June, one in July and one in August 2020). Figure 3a shows the plant development of one exemplary plot, which was surveyed throughout the measurement period. Different light conditions are noticeable, especially at the last three dates. As can be seen in Figure 3b, the Otsu threshold, which was applied to classify the DSM pixel either as 'plant' or as 'soil', shows good results under different light conditions but is challenged at the first two dates. However, according to Algorithm 1, the Otsu-segmentation is only relevant when outliers are present and $PH_{UAV}$ is set to $PH_{STD}$, which occurred for 14.51% of all plots at all dates (290 of 1999). Thus, Otsu-segmentation only affected plant height estimation in the minority of cases.

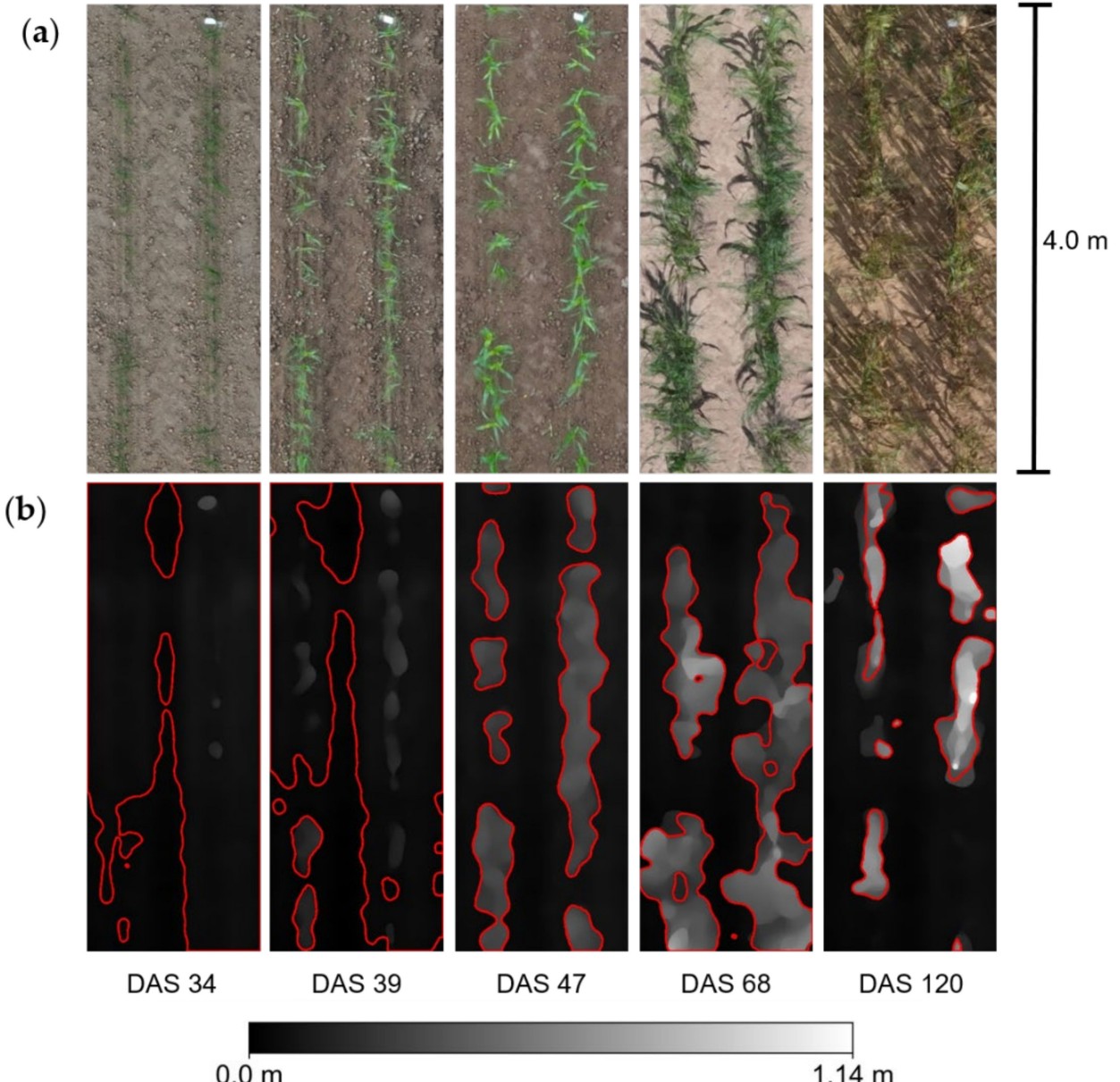

**Figure 3.** Maize plant development exemplified by one of the 400 genotypes surveyed by UAV at five stages during the growth period. (**a**) RGB image. The scale indicates the plot length; (**b**) heatmap of the digital surface model (DSM) for height estimation. The red contours indicate areas that were classified as plant and the grey scale bar shows plant height in meters; DAS: days after sowing.

The DSM in Figure 3b displays the plant development with up to 1.14 m plant height on the final date. The average plant height per measurement method and date as well as the respective standard deviations of all plots are shown in Figure 4. The standard deviations increase throughout the plant development and are most likely caused by the diversity of genotypes surveyed.

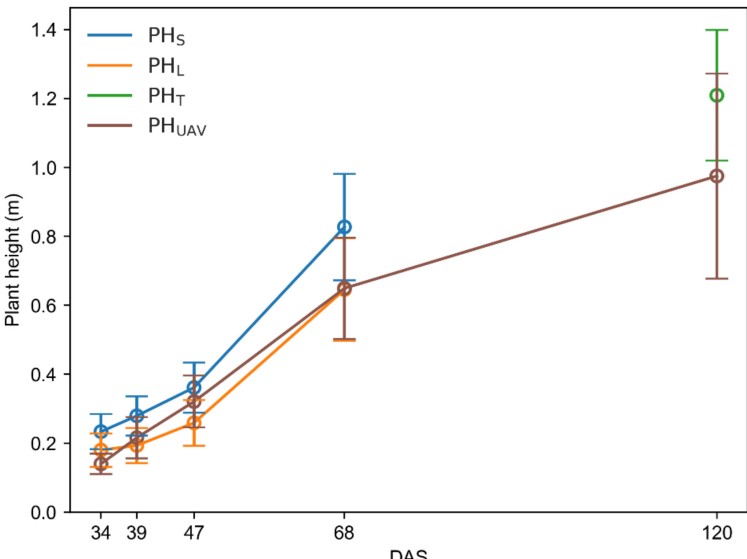

**Figure 4.** Average plant height of all plots during the growth period manually measured as distance to highest leaf ($PH_L$), highest straightened leaf ($PH_S$) and tassel tip ($PH_T$) as well as UAV-based estimated plant height ($PH_{UAV}$). Standard deviations are shown as error bars; DAS: days after sowing.

*3.2. Evaluation of Manual Measurement Methods and UAV-Based Plant Height Estimation*

First, the two manual methods of plant height measurement $PH_L$ and $PH_S$ were compared. Because leaves were straightened during measurement, the average $PH_S$ were higher than $PH_L$ on all dates. Plant height increased with parallel trend for both manual methods throughout the growth period and the standard deviations per date were similar (Figure 4). The ordinate intercepts of the regression lines as specified in Table 2 and shown in the left column of Figure 5 also demonstrate that $PH_S$ is generally higher than $PH_L$. Table 2 shows that the RMSE of $PH_L$ and $PH_S$ grew consistently from the first to the last date. As the leaf size increases throughout plant development, the growing RMSE of $PH_L$ and $PH_S$ was to be expected. The correlations between the manual methods were highly significant on all days ($p < 0.001$) and the $R^2$ was ranging from 0.47 to 0.62. Neither the $R^2$ nor the MAPE (30.90–48.61%) showed a clear trend related to the growth stage (Table 2).

**Table 2.** Predictive analytics of the height of the highest leaf ($PH_L$), the highest straightened leaf ($PH_S$), the tassel tip ($PH_T$) and the estimated plant height by UAV ($PH_{UAV}$); DAS: days after sowing, $r$: Pearson's correlation coefficient, RMSE: root mean square error, MAPE: mean absolute percentage error, $R^2$: coefficient of determination.

| Comparison | DAS | $r$ | RMSE (m) | MAPE (%) | Regression | $R^2$ |
|---|---|---|---|---|---|---|
| $PH_S$ vs. $PH_L$ | 34 | 0.70 * | 0.07 | 34.83 | y = 0.739x + 0.101 | 0.49 |
| | 39 | 0.75 * | 0.09 | 48.61 | y = 0.837x + 0.118 | 0.56 |
| | 47 | 0.68 * | 0.12 | 44.09 | y = 0.748x + 0.168 | 0.47 |
| | 68 | 0.79 * | 0.21 | 30.90 | y = 0.817x + 0.300 | 0.62 |
| $PH_{UAV}$ vs. $PH_L$ | 34 | 0.32 * | 0.06 | 25.48 | y = 0.197x + 0.104 | 0.10 |
| | 39 | 0.66 * | 0.05 | 22.87 | y = 0.774x + 0.066 | 0.44 |
| | 47 | 0.71 * | 0.08 | 28.89 | y = 0.804x + 0.113 | 0.50 |
| | 68 | 0.71 * | 0.11 | 14.00 | y = 0.701x + 0.197 | 0.51 |
| $PH_{UAV}$ vs. $PH_S$ | 34 | 0.42 * | 0.10 | 38.47 | y = 0.242x + 0.083 | 0.17 |
| | 39 | 0.73 * | 0.08 | 24.07 | y = 0.767x + 0.002 | 0.54 |
| | 47 | 0.78 * | 0.06 | 14.30 | y = 0.807x + 0.029 | 0.61 |
| | 68 | 0.70 * | 0.21 | 21.98 | y = 0.670x + 0.095 | 0.50 |
| $PH_{UAV}$ vs. $PH_T$ | 120 | 0.62 * | 0.33 | 21.74 | y = 0.974x − 0.203 | 0.38 |

\* $p < 0.001$.

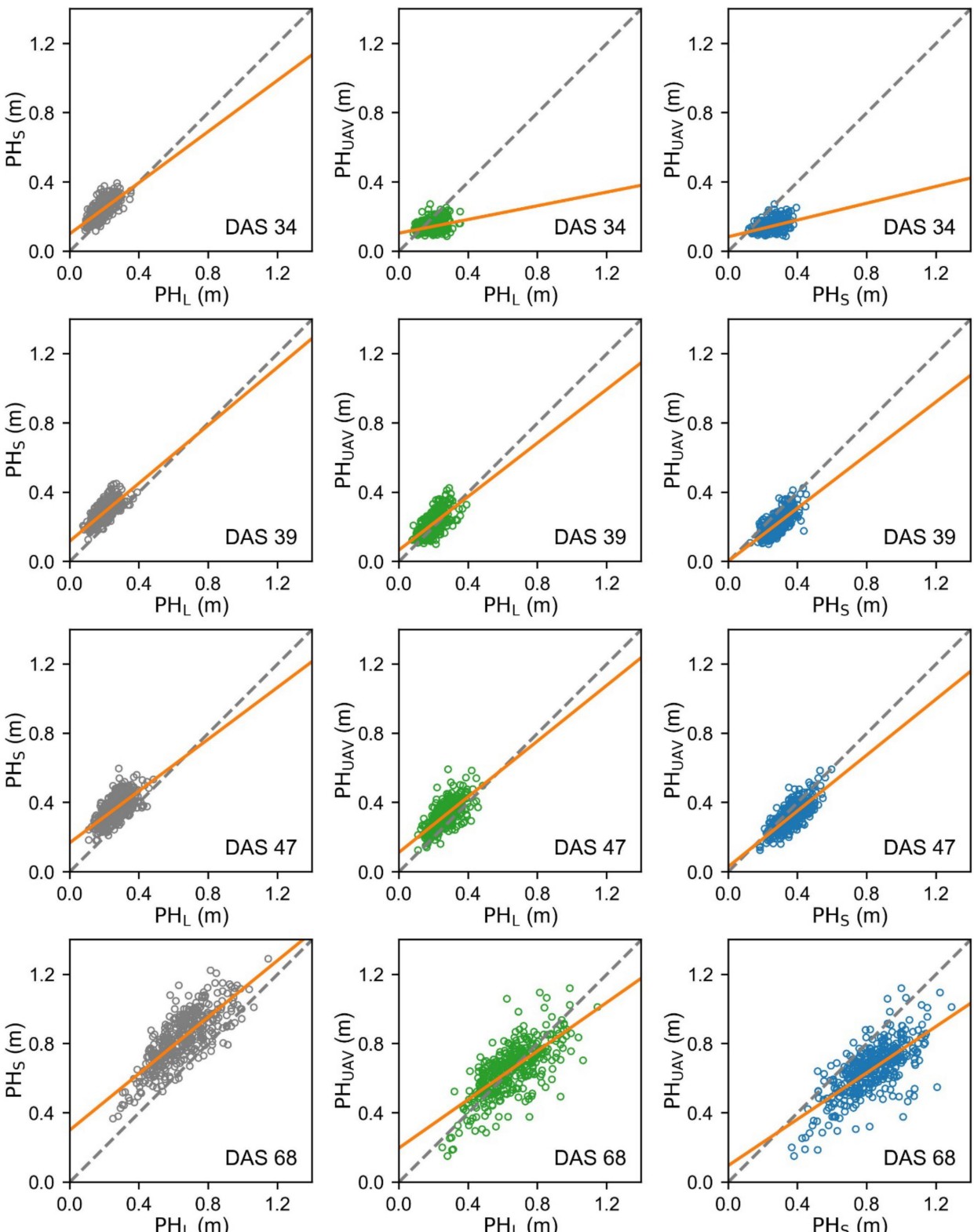

**Figure 5.** Manually measured plant height of the highest leaf ($PH_L$) and the highest straightened leaf ($PH_S$) compared to one another (grey) and UAV-based plant height estimation ($PH_{UAV}$) compared to $PH_L$ (green) and to $PH_S$ (blue). The linear regression line is shown in orange and its equation is specified in Table 2; DAS: days after sowing.

Since the maize plant did not appear to the UAV with straightened leaves but in its natural position, the mean of $PH_{UAV}$ was closer to the mean of $PH_L$ than to the mean of $PH_S$ across all plots on most dates (Figure 4). This was confirmed by comparing the RMSE between the manual methods and $PH_{UAV}$. Except for the flight on DAS 47, the RMSE of estimating $PH_S$ was always higher than that of estimating $PH_L$ (Table 2). DAS 47 was also the only date where the mean of $PH_{UAV}$ was closer to the mean of $PH_S$ than to the mean of $PH_L$ (Figure 4). Similarly, the MAPE between $PH_{UAV}$ and $PH_L$ was smaller than that of $PH_{UAV}$ and $PH_S$ on all days except for DAS 47 (Table 2).

In general, $PH_{UAV}$ showed a higher coefficient of determination to $PH_S$ than to $PH_L$, which were all based on highly significant correlations ($p < 0.001$) (Table 2). Only on DAS 68 the $R^2$ score of $PH_L$ (0.51) was slightly higher than that of $PH_S$ (0.50). According to the smaller RMSE and MAPE, $PH_{UAV}$ is the better direct estimate for $PH_L$ than for $PH_S$. Yet, the good correlation coefficients between $PH_{UAV}$ and $PH_S$ indicate potential for linear modeling between $PH_S$ and $PH_{UAV}$. The second and third column in Figure 5 show $PH_{UAV}$ of all plots in relation to the manual methods and their regression lines.

In contrast to the comparison of the manual methods, a clear trend can be observed for the MAPE and correlation coefficients between the manual methods and $PH_{UAV}$ over the growth period (Table 2). On DAS 34, the $R^2$ score between the manual methods (0.49) was clearly better than between $PH_{UAV}$ and $PH_L$ (0.10) or $PH_S$ (0.17). Nevertheless, from DAS 39 to DAS 68 the coefficients of determination between $PH_{UAV}$ and the manual methods are close to those of the manual methods. On DAS 47, $PH_L$ (0.50) and $PH_S$ (0.61) even showed a better $R^2$ score to $PH_{UAV}$ compared to the $R^2$ score between the manual methods (0.47) (Table 2).

The accuracy of estimating the height of the tassel tip on DAS 120 with MAPE = 21.74% and $R^2 = 0.38$ was rather poor (Table 2, Figure 6). Here, the standard deviation over all plots of $PH_{UAV}$ was clearly higher than that of $PH_T$ (Figure 4).

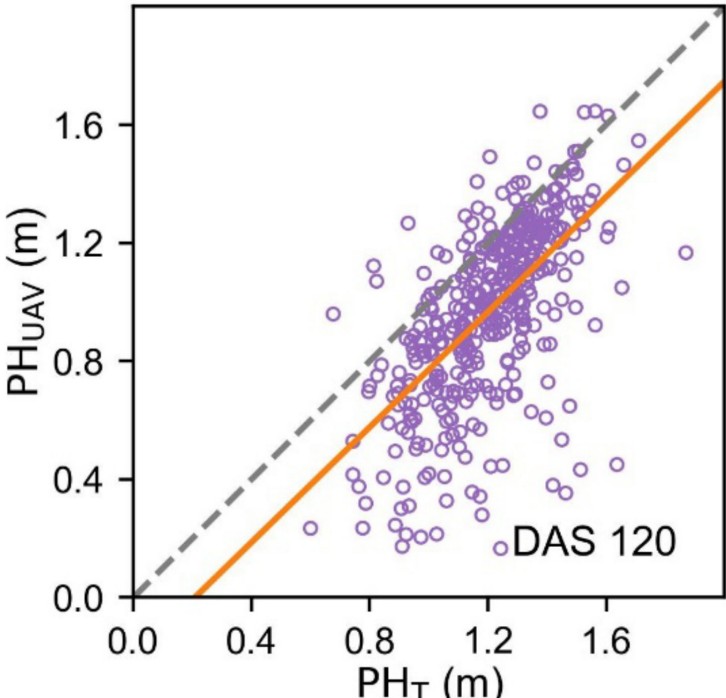

**Figure 6.** Comparison of UAV-based plant height estimation ($PH_{UAV}$) and manually measured tassel height ($PH_T$). The linear regression line is shown in orange and its equation is specified in Table 2; DAS: days after sowing.

How well the UAV-based plant height estimation performed was highly dependent on the plant developmental stage. On DAS 34, $PH_{UAV}$ was the least accurate for estimating both $PH_S$ (MAPE = 38.47%, $R^2$ = 0.17) and $PH_L$ (MAPE = 25.48%, $R^2$ = 0.10) (Table 2). Additionally, the standard deviation over all plots of $PH_{UAV}$ was clearly smaller than that of $PH_S$ and $PH_L$ on DAS 34, as can be seen in Figure 4. Promising results were obtained for $PH_S$ from DAS 39 to DAS 68 with 14.30–24.07% MAPE and 0.50–0.61 $R^2$. For estimating $PH_L$, likewise DAS 39 to DAS 68 was the best period with 14.00–28.89% MAPE and 0.44–0.51 $R^2$. The standard deviations on these dates were similar for all measuring methods (Figure 4).

### 3.3. Calibration of UAV-Based Plant Height Estimation

Although DAS 39 to DAS 68 was clearly the best period for UAV-based plant height estimation, the MAPE (14.00–28.89%) and RMSE (0.06–0.21 m) between $PH_{UAV}$ and the manual methods were still high. However, the similar $R^2$ values and regression equations on these dates (Table 2) highlight the potential for a standardized calibration of $PH_{UAV}$ for predicting $PH_L$ an $PH_S$ with separate models.

Thus, 279 (~70%) of the plots were randomly selected and their data of DAS 39 to DAS 68 was used to create linear regression models for calibration, which represent both manual methods (Figure 7a). By transforming the regression equations for $PH_L$

$$PH_{UAV} = 0.869 PH_L + 0.076 \tag{4}$$

and $PH_S$

$$PH_{UAV} = 0.746 PH_S + 0.031 \tag{5}$$

the calibration equations are

$$PH_{L,UAVcal} = \frac{PH_{UAV} - 0.076}{0.869} \tag{6}$$

and

$$PH_{S,UAVcal} = \frac{PH_{UAV} - 0.031}{0.746}. \tag{7}$$

**Table 3.** Predictive analytics of calibrated plant height estimated by UAV ($PH_{L,\,UAVcal}$, $PH_{S,\,UAVcal}$) vs. measurement of highest leaf ($PH_L$) and highest straightened leaf ($PH_S$); DAS: Days after sowing, $r$: Pearson's correlation coefficient, RMSE: Root mean square error, MAPE: Mean absolute percentage error, $R^2$: Coefficient of determination.

| Comparison | DAS | $r$ | RMSE (m) | MAPE (%) | Regression | $R^2$ |
|---|---|---|---|---|---|---|
| $PH_{L,\,UAVcal}$ vs. $PH_L$ | 39 | 0.69 * | 0.07 | 28.39 | y = 1.043x − 0.049 | 0.48 |
| | 47 | 0.73 * | 0.06 | 18.37 | y = 0.930x + 0.039 | 0.54 |
| | 68 | 0.66 * | 0.14 | 17.49 | y = 0.743x + 0.193 | 0.43 |
| $PH_{S,\,UAVcal}$ vs. $PH_S$ | 39 | 0.80 * | 0.06 | 19.62 | y = 1.142x − 0.077 | 0.64 |
| | 47 | 0.78 * | 0.07 | 14.15 | y = 1.090x − 0.008 | 0.61 |
| | 68 | 0.75 * | 0.13 | 12.07 | y = 0.981x + 0.017 | 0.56 |

* $p < 0.001$.

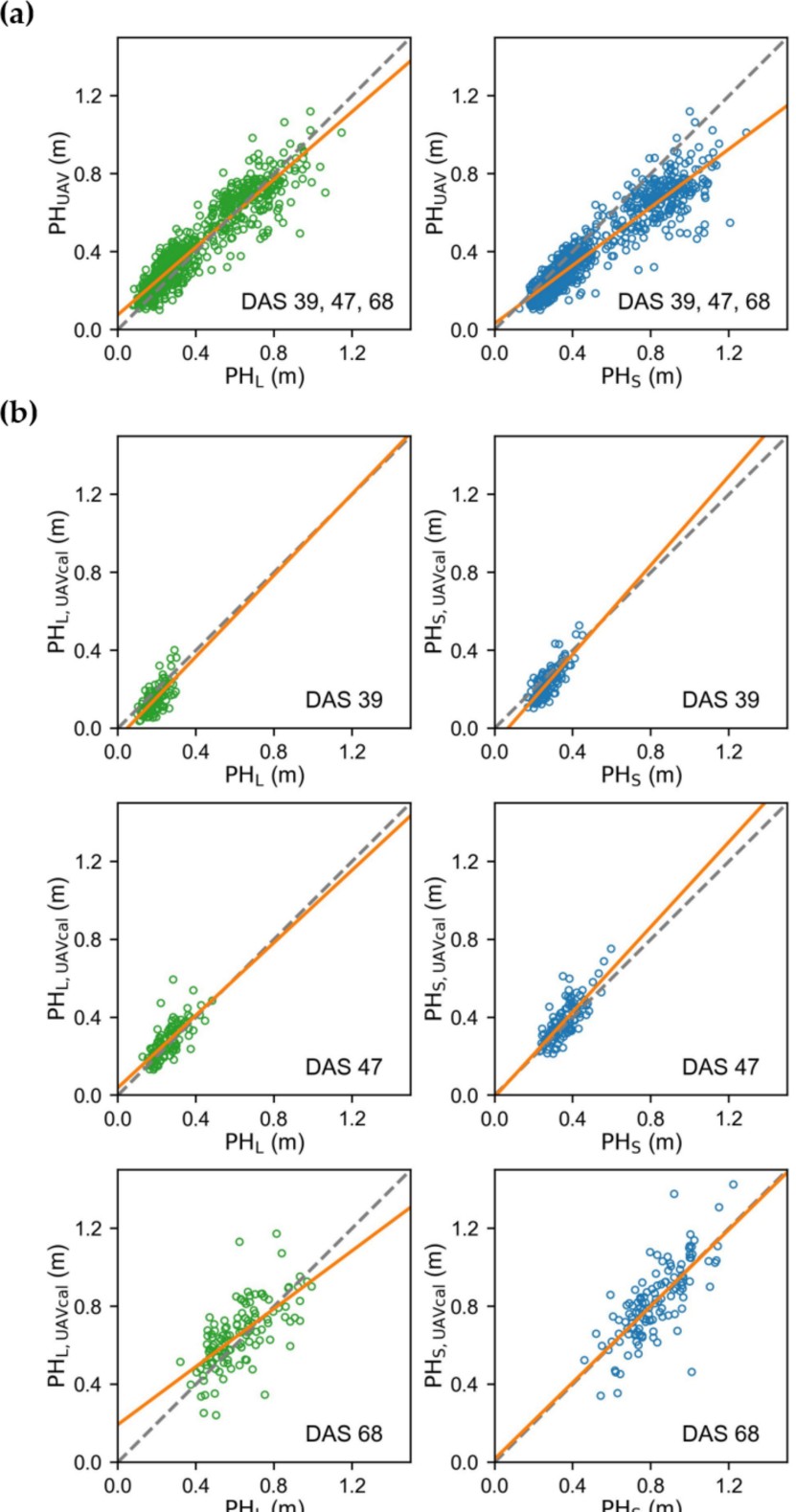

**Figure 7.** Calibration of UAV-based plant height estimation (PH$_{UAV}$) based on highest leaf (PH$_L$) and highest straightened leaf method (PH$_S$). The linear regression lines are shown in orange. (**a**) Data set for calibration of three dates combined. (**b**) Validation of calibrated data for predicting the plant height of highest leaf (PH$_{L, UAVcal}$) and the plant height of the highest straightened leaf (PH$_{S, UAVcal}$) based on UAV measurements. The regression equations are specified in Table 3; DAS: days after sowing.

The calibration models were applied to the remaining 120 plots (~30% of all plots) for each date separately (Figure 7b). The resulting RMSE and MAPE were slightly worse for predicting $PH_L$ after calibration (0.06 m $\leq$ RMSE $\leq$ 0.14 m; 17.49% $\leq$ MAPE $\leq$ 28.39%) (Table 3) than before (0.05 m $\leq$ RMSE $\leq$ 0.11 m; 14.00% $\leq$ MAPE $\leq$ 28.89%) (Table 2). However, the regression lines in the left column of Figure 7b all show a slope close to 1 and a relatively small intercept compared to Figure 5, which indicates that using a standardized calibration model trained on data of three dates is generalizing well and is feasible to be applied on single dates. Since the calibrated $PH_{L, UAVcal}$ does not show a lower RMSE and MAPE, there does not seem to be a systematic error of $PH_{UAV}$ for estimating $PH_L$. In contrast to $PH_{L, UAVcal}$, $PH_{S, UAVcal}$ showed a lower RMSE and MAPE (0.06 m $\leq$ RMSE $\leq$ 0.13 m; 12.07% $\leq$ MAPE $\leq$ 19.62%) (Table 3) than $PH_{UAV}$ (0.06 m $\leq$ RMSE $\leq$ 0.21 m; 14.30% $\leq$ MAPE $\leq$ 38.47%) (Table 2) for estimating $PH_S$. Thus, a systematic error was present, which was caused by straightening the leaf for manual measurement. The decreased errors demonstrate that the calibration diminishes the error. The regression lines between $PH_{S, UAVcal}$ and $PH_S$ again show slopes close to 1 and intercepts close to 0 at all dates (Figure 7b, left column; Table 3), indicating that the calibration model was generalizing well again.

## 4. Discussion

In this study, we demonstrated that the accuracy of estimating plant height in maize by a UAV is highly dependent on the plant growth stage but also on the manual measurement method to which $PH_{UAV}$ is compared. Accuracy was especially poor on DAS 34 ($PH_S$: $R^2 = 0.17$, $PH_L$: $R^2 = 0.10$) and on DAS 120 ($R^2 = 0.38$) (Table 2). The low accuracy on DAS 34 was most likely due to the still small plant sizes on that date. The average value of $PH_L$ on DAS 34 was 0.18 m (Figure 4), which could be missed in the range of soil coarseness. As can be seen in Figure 3b, on DAS 34 the young plants could be barely recognized on the DSM, which held true for most of the plots. Accordingly, the standard deviation of $PH_{UAV}$ over all plots was comparably small (Figure 4). However, on DAS 39 the average $PH_L$ (0.19 m) was barely higher than on DAS 34 (0.18 m) (Figure 4) yet the accuracy had clearly increased ($R^2 = 0.44$) (Table 2). Thus, the DSM quality might not only be influenced by plant height but also by plant morphology. Nevertheless, an average $PH_L$ of 0.2 m should be sufficient for the presented approach to achieve reasonable results. However, different light conditions, as can be seen on DAS 47 and 68 (Figure 3a), do not seem to affect the estimation accuracies seriously (Table 2).

On DAS 120 many maize genotypes already reached physiological maturity and the leaves were senescing (Figure 3a), hampering the detection by the DSM. As some genotypes did not reach maturity on DAS 120 but others did, the standard deviation of $PH_{UAV}$ on that date is clearly higher than that of $PH_T$ (Figure 4). From DAS 39 to DAS 68 the highest accuracies were achieved with up to $R^2 = 0.61$ compared to $PH_S$ (Table 2). Accordingly, this period is the optimal period to estimate maize plant height with the presented approach.

The data of this period was used to train one calibration model each for predicting $PH_L$ and $PH_S$. As $PH_L$ is measured without changing the plants' appearance there is no systematic error to be expected when compared to $PH_{UAV}$. Consequently, the calibration did not improve the estimation. However, the MAPE and RMSE of estimating $PH_S$ from calibrated $PH_{UAV}$ was considerably improved. Since the leaves were straightened for the manual measurement of $PH_S$, it was to be expected that calibration will diminish this systematic error. Different plots were used for training and validation of the calibration model and these plots represented random genotypes. Moreover, no systematic error could be shown for $PH_{L, UAVcal}$. Thus, the shown systematic error of $PH_{S, UAVcal}$ is independent of the maize genotype and the UAV platform but instead is related to the manual method $PH_S$. The MAPE between $PH_{S, UAVcal}$ and $PH_S$ (12.07% $\leq$ MAPE $\leq$ 19.62%) are the smallest of all methods compared. Accordingly, by applying a linear regression model for calibration, $PH_{UAV}$ is the better value for predicting $PH_S$ than for $PH_L$ although only leaves in their natural position are perceived by the UAV.

In contrast to $PH_L$, where only a small proportion of the highest leaf influences the measured height, $PH_S$ captures the whole leaf length of the highest straightened leaf. Thus, $PH_S$ might be the better indicator for biomass since the whole leaf length is represented. The poor results on DAS 34 and the better results afterwards indicate that a higher green biomass facilitates UAV-based height measurement. By capturing information about biomass, the estimation of $PH_S$ might be more interesting than that of $PH_L$ from a breeder's perspective.

Su et al. [9] ($R^2 = 0.78$) and Gilliot et al. [10] ($R^2 = 0.9$) achieved higher accuracies than our approach in estimating plant height of maize based on DSM, which is most likely due to the different methodologies. While we compared $PH_{UAV}$ to averaged manual values per plot, Su et al. [9] and Gilliot et al. [10] both compared UAV based estimations to individual plants tracked by GPS. Thereby the manual error of selecting representative plants is excluded completely. Watanabe et al. [13] followed a similar approach to ours in estimating sorghum height based on DSM. They manually measured two plants per plot and compared the averaged value to the UAV-based estimation of the center of the plot and achieved a correlation coefficient of up to $r = 0.52$. In this study, $PH_{UAV}$ of all dates, except the first one, scored a correlation coefficient higher than $r = 0.52$ when compared to the different manual measurement methods (Table 2).

Furthermore, Watanabe et al. [13] pointed out that different types of errors of manual and UAV measurements impair each other's estimation. The correlation coefficients in this study indicate that the relationship of $PH_{UAV}$ to the manual methods is as close as the manual methods are to each other (Table 2), which indicates a non-systemic error between the manual methods. It is questionable whether manual measurements in maize by three plants per plot represent the ground truth sufficiently. Especially the measurement of tall maize genotypes and the subjective decision to select three representative plants, as performed in this study, bear a high risk of bias. However, UAV-platforms are more affected by environmental conditions than manual workers and cannot produce reliable results under weather conditions such as, for instance, strong winds and rain.

## 5. Conclusions

In this study, the feasibility of UAV-based plant height estimation at different growth stages of maize was evaluated. After obtaining the best results for estimating both $PH_L$ ($0.44 \leq R^2 \leq 0.51$) and $PH_S$ ($0.50 \leq R^2 \leq 0.61$) from 39 to 68 days after sowing (DAS), it can be recommended to apply the approach presented here from 0.2 m average plant height to maturity before the plants start to senesce and change the leaf color. The preferred method of plant breeders to measure plant height, namely from ground to the tip of the straightened highest leaf, can be estimated based on UAV measurements after calibration with a MAPE in a range from 12.07% to 19.62%. In future research, accuracy of DSM-based plant height estimation, especially after plant senescing, could be improved by capturing biomass per plant, potentially by means of deep learning-based computer vision. UAV is an efficient and promising platform in plant phenotyping and holds the potential to reliably automatize labor-intense measurements and thus save precious time in plant breeding.

**Author Contributions:** Conceptualization, A.-J.R. and J.M.; methodology, L.H.O. and A.-J.R.; software, L.H.O. and A.-J.R.; validation, L.H.O. and A.-J.R.; formal analysis, L.H.O. and A.-J.R.; investigation, A.-J.R. and T.M.W.; resources, L.H.O., A.-J.R., T.M.W., T.W. and J.M.; data curation, T.M.W. and A.-J.R.; writing—original draft preparation, L.H.O. and A.-J.R.; writing—review and editing, T.W., T.M.W., X.H. and J.M.; visualization, L.H.O. and A.-J.R.; supervision, T.W. and J.M.; project administration, T.W., X.H. and J.M.; funding acquisition, T.W., X.H. and J.M. All authors have read and agreed to the published version of the manuscript.

**Funding:** This research was funded by the Deutsche Forschungsgemeinschaft (DFG, German Research Foundation)—328017493/GRK 2366 (Sino-German International Research Training Group AMAIZE-P).

**Data Availability Statement:** The data presented in this study are available on request from the corresponding author. The data are not publicly available due to ongoing research.

**Acknowledgments:** The authors thank Mire Halilaj, Jochen Jesse, Franz Josef Mauch, Heralt Pöschel, Benjamin Schleicher and Clara Bartenstein from the institute of plant breeding for performing the plant height measurements. Furthermore, we want to acknowledge Sabine Nugent for language editing of this manuscript.

**Conflicts of Interest:** The authors declare no conflict of interest.

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
