# Peer review of "Remote Sensing of Maize Plant Height at Different Growth Stages Using UAV-Based Digital Surface Models (DSM)"

_agronomy, doi:10.3390/agronomy12040958_

Round 1

Reviewer 1 Report

Amend the manuscript based on the following comments :

Mention UAV acquisition and ground truth measurement details observed for the experiment  mentioning UAV Flight and Ground Measurement Dates,Flight Time, Illumination (lux) ,Sky Condition ,Wind (ms−1) etc.

Include field experiment overview figure.

Where is the flowchart depicting the general overview of the methodology for calculating the plant height?.

Mention in detail about the procedure for generation of the digital surface model. Comprehensive detailing is needed for clarity.

Include equations in statistical analysis to assess 3 metrics
 like coefficient of determination (R2), mean absolute error (MAE), and root mean square error (RMSE)—for maize plant height at different growth satges to evaluate the strength of the relationship between plant heights.

Reviewer 2 Report

Colleagues! A ruler of detail is needed, depending on the scale of the survey. Also an assessment of the influence of meteorological data, for example, restrictions on wind load (wind speed) when using the method.

Reviewer 3 Report

The objectives, methods, research and conclusions of the research are explained very clearly and with relevant detail.  The authors addressed some issues, including pros and cons related to averaging measurments across various genotypes, and discuss some examples of using GPS plants to compare in situ  vs remotely sensed measurements.  However, I think it would be interesting to pull out data from a few dominant varieties in a classified comparison.  Perhaps in a subsequent paper.

Reviewer 4 Report

This paper deals with the advantages and limitations of the modern technologies (unmanned aerial vehicles) in the estimations of plant traits that are important in breeding programs (plant height) and that are extremely labor demanding and time consuming when performed in the usual way. This interesting and practically applicable work is within the Aims and Scope of the journal. The article is concisely and clearly written. I suggest to accept it for publication, after a minor correction regarding terminology.

Reviewer 5 Report

The proposed manuscript addresses the use of am automated methodology, based on UAV collected images, to assess maize height throughout its growth and compare to a manual methodology. This approach has its merits, and the manuscript is well structured and written in good English. However, a few considerations and remarks (next presented) should be raised, which need further clarification before this manuscript can be considered for publication. Hence, I believe that the proposed manuscript needs major revision before being further assessed.

  • In the Abstract section, line 27-29, the authors state “It is recommended to apply UAV-based maize height estimation from 0.2 m average plant height to maturity before the plants start to senesce and change the leaf color.” If senescence determination is one of the primordial goals, why not use leaf color to estimate maturity?
  • In the Introduction section, line 46-47, what are the main advantages of the proposed methodology regarding the ones presented in the 6-11 references?
  • In the Introduction section, line 65-67, the authors state “In particular, our objectives were to (i) identify the optimal period for UAV-based plant height estimation”. Why? Shouldn’t be the primordial objective to estimate the maize height when is mature (before senescence)?
  • In the Materials and Methods section, line 83, and as referred in the Discussion, why use only three plants for the manual measurements? This seems rather low for me and a drawback regarding the accuracy of the manual measurements. Furthermore, how many plants were surveyed by the UAV methodology for each date and method?
  • In the Materials and Methods section, regarding the Procedure of plant height estimation, what was the soil metric area for the computation of the minimum, maximum and mean plant and soil values? What was the maximum altitude difference in the soil?
  • In the Materials and Methods section, again regarding the Procedure of plant height estimation, and taking into consideration the results obtained in Figure 1, have the authors tried to use other segmentation algorithms besides Otsu (other histogram based methods, entropy based, neural networks, etc)? Or even perform manual segmentation? Otsu algorithm limitations are well known.
  • In the Materials and Methods section, regarding Algorithm 1, what are the basis for adding the value of five times the standard deviation to the average plant height and subtracting the value of five times the standard deviation of the soil average height, and then removing the later from the first in estimating the plant height?
  • In Figure 1, in DAS 34 and 39 the distinction between soil and plant seems rather poor to me and a serious limitation to the obtention of good results for these days.
  • In the Results section, line 236-237, the authors state “Good results were obtained for PHS from DAS 39 to DAS 68 with 14.30 % – 24.07 % MAPE and 0.50 – 0.61 R2.”. For R2 ranging from 0.5 to 0.61 I would use the term "Reasonable" or "Promising" rather than "Good". The same with respect to line 303 in the Discussion section.
  • In the Results section, line 245, please rephrase “…the manual methods were still on a high level.” to “…the manual methods were still high.”.
  • In the Results section, regarding the Calibration of UAV-based plant height estimation, please indicate the obtained linear regression models corresponding to figure 5a before presenting the calibration equations (though it can be inferred from these equations). Also, please number the presented equations.
  • In the Discussion section, shouldn’t DAS 120 be the most important day given its closeness to the maturation stage and, therefore, be the standard for the analysis of the goodness of fit of the proposed methodology?
  • In the Discussion section, line 294-295, the authors state “The low accuracy on DAS 34 was most likely due to the still small plant sizes on that date.”. What about the poor recognition ability of the performed image processing (Otsu segmentation) of the collected images?
  • In the Discussion section, line 309-310, the authors state “The height of green plants was depicted more accurately than that of brown plants as can be seen on DAS 120 in Figure 1b”. However, It's hard to take that conclusion from the presented figure (not enough color detail).
  • In the Discussion section, line 312-313, the authors refer to the period between DAS 39 to 68 as “the optimal period to estimate maize plant height with the presented approach.”. Again, shouldn’t the primordial goal of the presented methodology to help determine the period in which the plant reaches its highest size before senescence? I believe that the determination of an "optimal period to estimate plant height" is not that significant. Indeed, according to the results, the developed methodology is only suitable for estimating plant height up until 50 days before reaching the maximum height (solely slightly above mid growth period).
  • In the Discussion section, line 328-329, the authors state “PHS is the manual method that was estimated with the highest accuracy in this study, which might have been due to indirect capturing of biomass.”. I don't understand what is intended to state here.
  • In the Discussion section, line 332-334, the authors state “The poor results on DAS 34 and the better results afterwards indicate that a higher biomass facilitates UAV-based height measurement.”. However, at DAS 120 the biomass was higher and the recognition was lower.
  • In the Discussion section, line 348-351, the authors state “The correlation coefficients in this study indicate that the relationship of PHUAV to the manual methods is as close as the manual methods are to each other (Table 1), which demonstrates that the presented approach provides as good results as the manual measurement methods.”, and again on the Conclusions section, line 362-365, “The feasibility of UAV-based plant height estimation is furthermore underlined by the manual methods PHL and PHS (0.47 ≤ R2 ≤ 0.62) being as closely related to one another as PHUAV to the manual methods.”. I don't understand what the authors intend to state here. I believe that the authors cannot compare the difference of two manual methods that measure different things (TS and TL) with the difference between the automatic method for a given manual method. The comparison should only be performed between the TS, or the TL, mean and standard deviation with the corresponding estimated values by the automatic method, and never between TS and TL.

Round 2

Reviewer 5 Report

The authors have successfully answered the points raised in the first review. Therefore, I now believe that the proposed manuscript can be accepted in the present form.